# Status of zoonotic disease research in refugees, asylum seekers and internally displaced people, globally: A scoping review of forty clinically important zoonotic pathogens

Regina Oakley[1,2]*, Nadja Hedrich[3], Alexandra Walker[1,2], Habtamu Merha Dinkita[4], Rea Tschopp[2,4,5], Charles Abongomera[1,2], Daniel H. Paris[1,2]

**1** Department of Medicine, Swiss Tropical and Public Health Institute, Allschwil, Switzerland, **2** University of Basel, Basel, Switzerland, **3** Epidemiology, Biostatistics and Prevention Institute, University of Zürich, Zürich, Switzerland, **4** One Health Division, Armauer Hansen Research Institute, Addis Ababa, Ethiopia, **5** Department of Epidemiology and Public Health, Swiss Tropical and Public Health Institute, Allschwil, Switzerland

* regina.oakley@swisstph.ch

**Data Availability Statement:** Data analyzed in this study is available in S12 Table.

## Abstract

### Background

At the end of 2022, there were over 108 million forcibly displaced people globally, including refugees, asylum seekers (AS) and internally displaced people (IDPs). Forced migration increases the risk of infectious disease transmission, and zoonotic pathogens account for 61% of emerging and re-emerging infectious diseases. Zoonoses create a high burden of disease and have the potential to cause large-scale outbreaks. This scoping review aimed to assess the state of research on a range of clinically relevant zoonotic pathogens in displaced populations in order to identify the gaps in literature and guide future research.

### Methodology / Principal findings

Literature was systematically searched to identify original research related to 40 selected zoonotic pathogens of interest in refugees, AS and IDPs. We included only peer-reviewed original research in English, with no publication date restrictions. Demographic data, migration pathways, health factors, associated outbreaks, predictive factors and preventative measures were extracted and synthesized. We identified 4,295 articles, of which 347 were included; dates of publications ranged from 1937 to 2022. Refugees were the most common population investigated (75%). Migration pathways of displaced populations increased over time towards a more complex web, involving migration in dual directions. The most frequent pathogen investigated was *Schistosoma* spp. (n = 99 articles). Disease outbreaks were reported in 46 publications (13.3%), with viruses being the most commonly reported pathogen type. Limited access to hygiene/sanitation, crowding and refugee status were the most commonly discussed predictors of infection. Vaccination/prophylaxis drug administration, surveillance/screening and improved hygiene/sanitation were the most commonly discussed preventative measures.

**Funding:** This study was funded by the Stanley Thomas Johnson Foundation (Grant number 1053-KF; awarded to DHP; https://www.johnsonstiftung.ch/). The funders had no role in study design, data collection and analysis, decision to publish, or preparation of the manuscript.

**Competing interests:** The authors have declared that no competing interests exist.

## Conclusions / Significance

The current research on zoonoses in displaced populations displays gaps in the spectrum of pathogens studied, as well as in the (sub)populations investigated. Future studies should be more inclusive of One Health approaches to adequately investigate the impact of zoonotic pathogens and identify transmission pathways as a basis for designing interventions for displaced populations.

### Author summary

Currently, there are over 108 million forcibly displaced people globally—mainly refugees, asylum seekers and internally displaced people. Migration is associated with increased exposure to infectious diseases. Zoonotic pathogens transmitted between humans and animals, account for the majority of emerging human diseases. Zoonoses are responsible for a substantial disease burden and are recognised to cause disease outbreaks, particularly in vulnerable populations in crowded living conditions. Unfortunately, zoonoses are often neglected by both healthcare services, medical research and especially in the context of migration. This review focussed on the current research of a spectrum of clinically relevant zoonotic pathogens in displaced populations to identify gaps in literature and inform on future research.

Our findings highlighted the following gaps; zoonotic viruses appear particularly neglected, despite them being most commonly associated with disease outbreaks. The majority of publications investigating zoonoses in displaced people focused on three pathogens: *Schistosoma* species, *Giardia lamblia* and *Leishmania* species. There were very few publications investigating zoonotic pathogens in asylum seekers and internally displaced people compared to refugees. Future studies should be more inclusive and incorporate One Health approaches to adequately investigate the impact of zoonoses on displaced populations, and to identify transmission pathways for designing improved interventions.

## Introduction

Globally, there were over 108 million forcibly displaced people as of the end of 2022, including 35.3 million refugees, 62.5 million internally displaced people (IDP), and 5.4 million asylum seekers (AS) [1]. Although the terms refugee and AS are often used interchangeably, they are distinct groups which can face specific challenges.

The United Nations defines refugees as any person who: (1) Has been considered a refugee under the currently valid Arrangements or Conventions or the Constitution of the International Refugee Organization [2,3]; or (2) ". . .owing to well-founded fear of being persecuted for reasons of race, religion, nationality, membership of a particular social group or political opinion, is outside the country of his nationality and is unable or, owing to such fear, is unwilling to avail himself of the protection of that country; or who, not having a nationality and being outside the country of his former habitual residence as a result of such events, is unable or, owing to such fear, is unwilling to return to it. In the case of a person who has more than one nationality, the term "the country of his nationality" shall mean each of the countries of which he is a national, and a person shall not be deemed to be lacking the protection of the

country of his nationality if, without any valid reason based on well-founded fear, he has not availed himself of the protection of one of the countries of which he is a national" [3].

While, <u>AS</u> are defined as: "A person who has left his/her country of origin and formally applied for asylum in another country but whose claim has not yet been concluded. If an applicant is denied refugee status at the first instance level during refugee status determination, he/she can appeal this negative decision. Until a final decision is reached, this person remains an asylum-seeker." [4].

<u>IDPs</u> on the other hand are defined as: "Persons or groups of persons who have been forced or obliged to flee or to leave their homes or places of habitual residence, in particular as a result of or in order to avoid the effects of armed conflict, situations of generalized violence, violations of human rights or natural or human-made disasters, and who have not crossed an internationally recognized State border" [5].

Seventy-six percent of the forcibly displaced people at the end of 2022 were hosted in low- and middle-income countries (LMICs), while 20% were hosted by the least developed countries, increasing the pressure on already over-burdened health systems [1,6]. Furthermore, LMICs often have limited diagnostic capacity or comprehensive surveillance systems [7]. In low and middle income settings, endemic zoonotic diseases are responsible for an estimated 20% of infectious disease burden in humans, and due to limited evidence, this might represent an under estimation [8,9].

Certain zoonotic diseases are related to direct transmission from animals, such as brucellosis and rabies [7]. However, others originate at the animal-human interface and may initiate an outbreak with rapid spread by human-to-human transmission [7]. This second scenario is often associated with large scale epidemics or pandemics, as in the case of Human Immunodeficiency Virus (HIV) or more recently the severe acute respiratory syndrome coronavirus 2 (SARS-CoV-2) pandemic [7,10].

Factors that determine the capacity for pathogens to spread include: high population density (e.g. population growth, urbanization, crowded settings–prisons, slums or camps for displaced people), increased mobility (e.g. migration and travel), changes in social structure (e.g. aging population, increased numbers of immunocompromised people, maternal employment leading to higher childcare centre use) or human behaviour (e.g. food and water preparation, use of mosquito nets, illicit drug use, unprotected sex), and the breakdown in health infrastructure (e.g. conflict) [7,11]. These factors are common among displaced populations, who are often housed in overcrowded and unhygienic camps that favour disease transmission [12,13]. Forced migration is a recognized factor leading to infectious disease emergence, with zoonotic diseases accounting for 61% of all emerging and re-emerging human pathogens, however, they are often neglected by both health services and research [14,15]. Zoonotic diseases account for 15 of the 20 neglected tropical disease conditions listed by the World Health Organization [16].

The overall aim of this review was to assess the state of research on a panel of clinically relevant zoonotic pathogens in refugees, IDPs and AS, globally. The spectrum of selected pathogens (n = 40) was compiled in consultation with physicians, veterinarians and researchers having expertise in tropical medicine, infectious zoonotic diseases and causes-of-febrile illness studies. A range of zoonotic pathogens representing clinically important agents, including viruses, bacteria, protozoa and helminths was selected to identify trends and evidence in zoonotic disease research within displaced populations. There are well over 800 known zoonotic human-pathogenic agents globally, and the spectrum of clinically relevant pathogens included in this review had to be limited to align with the feasibility of developing a database search string that accurately included all search terms for all included zoonotic pathogens and their diseases [14].

The review aimed to address the following research questions, and to identify gaps in the current published literature on zoonoses in displaced populations in order to guide future research. Which of the pathogens of interest are: i) most frequently investigated in these populations?; ii) associated with migration routes of displaced people?; iii) commonly reported from Africa, Asia and Latin America?; iv) associated with outbreaks in displaced people, globally?; and what are the most commonly reported predictors and preventative measures associated with zoonotic diseases in displaced populations, globally?

## Materials and methods

### Eligibility criteria

The review included studies on refugees, AS and IDPs that investigated at least one of the 40 zoonotic pathogens of interest (Table 1). Only publications with a clinical or laboratory diagnosis of one of the pathogens were included. Publications describing self-reported cases were excluded. Only peer-reviewed original research in English was included. No limitation was placed on publication date. Publications where full texts could not be obtained were excluded.

### Data search strategy

A literature search was conducted in the PubMed, Embase, Cochrane library, Scopus and Web of Science databases. The search included 40 clinically relevant zoonotic pathogens, comprised of bacteria, viruses and parasites (Table 1). The 'displaced people', 'zoonosis' and 'disease specific name' strings were developed in PubMed and translated using the Polyglot tool from Bond University [17]. The COVID-19 search strings were adapted from the Canadian Agency for drugs and Technology in Health for PubMed, EMBASE and Scopus [18]. 'Supplementary concept' terms were updated to the MeSH term where available in PubMed. The translated COVID-19 search string from PubMed was used for Web of Science and the Cochrane library. MeSH terms were replaced by Emtree terms in the EMBASE translation. Duplicate search terms, generated in the Web of Science translation due to the lack of the MeSH term system, were removed. Complete search strings were reviewed by an information specialist at the University Medical Library, University of Basel (Basel, Switzerland). The full search strategy is available in the supplementary material (S2 Text). An initial search was conducted on the 5th May 2021 and the final search on the 26th December 2022. A snowballing approach was used to identify additional articles by searching the reference lists of retrieved reviews.

### Screening and selection

Records identified by our search strategy were imported to EndNote X9 (Clarivate Analytics) where duplicates were identified and removed. Records were independently screened for eligibility criteria by two reviewers (RO, and either NH or HM) using Rayyan [19]. Conflict between article selections was resolved through discussion between the two reviewers (RO, and either NH or HM; and/or discussion with RT or DP). Two reviewers (RO, and either NH or HM) similarly screened full text articles. Full texts were obtained through the electronic databases, the Basel University library and direct contact with authors.

### Data extraction and synthesis

A data extraction tool was developed in Microsoft Excel and pilot tested by two reviewers (RO and HM) extracting 20 publications independently to determine consistency between reviewers. Data extraction of the remaining articles was performed by four reviewers (RO, NH, AW and CA). Ten percent of the publications were extracted in duplicate, independently to check

**Table 1. Zoonotic pathogens of interest included in the scoping review.**

| Pathogen | Disease |
| --- | --- |
| **Bacteria** | |
| *Anaplasma* spp. | Anaplasmosis |
| *Borrelia recurrentis* | Relapsing Fever |
| *Brucella* spp. | Brucellosis |
| *Coxiella burnetii* | Q Fever |
| *Francisella tularensis* | Tularaemia |
| *Leptospira* spp. | Leptospirosis |
| *Orientia tsutsugamushi* | Scrub typhus |
| *Rickettsia* spp. | Rickettsiosis, Spotted fever, Epidemic typhus, Murine/ Endemic typhus |
| *Salmonella enterica* subsp. *enterica* serovar typhi | Typhoid Fever |
| *Yersinia pestis* | Plague |
| **Parasites** | |
| *Cryptosporidia* spp. | Cryptosporidiosis |
| *Echinococcus* spp. | Echinococcosis |
| *Fasciola* spp. | Fascioliasis |
| *Filarioidea* | Filariasis |
| *Giardia lamblia* | Giardiasis |
| *Leishmania* spp. | Cutaneous leishmaniasis, Visceral leishmaniasis, Mucosal leishmaniasis |
| *Schistosoma* spp. | Schistosomiasis |
| *Taenia solium* | Taeniasis, cysticercosis |
| *Toxoplasma gondii* | Toxoplasmosis |
| *Trichinella* spp. | Trichinellosis/Trichinosis |
| *Trypanosoma* spp. | African sleeping sickness, Chagas disease |
| **Viruses** | |
| Chikungunya virus | Chikungunya fever |
| Crimean-Congo haemorrhagic fever virus | Crimean-Congo haemorrhagic fever |
| Dengue virus | Dengue fever |
| Ebola virus | Ebola haemorrhagic fever |
| Hantavirus | Hantavirus pulmonary syndrome |
| Hepatitis E virus | Hepatitis |
| Japanese encephalitis virus | Japanese encephalitis |
| Lassa virus | Lassa haemorrhagic fever |
| Marburg virus | Marburg virus disease (haemorrhagic fever) |
| MERS CoV | Middle East respiratory syndrome coronavirus |
| Nipah virus | Nipah virus infection |
| Rabies virus | Rabies |
| Rift valley fever virus | Rift Valley Fever |
| SARS CoV | Severe acute respiratory syndrome coronavirus |
| SARS CoV-2 | Coronavirus disease (COVID-19) |
| Tick-borne encephalitis virus | Tick-borne encephalitis |
| West Nile virus | West Nile fever |
| Yellow fever virus | Yellow fever |
| Zika virus | Zika fever |

that consistency between reviewers was maintained. The data extracted included study type, demographics (population, setting), migration (origin country/region, host country/region, transit country/region), health (pathogen of interest, sampling year, case type (outbreak, individual, screening), other health concerns, preventative measures and disease predictors). Migration regions were assigned based on the United Nations (UN) Geoscheme, which reflects homogeneity of populations and demographic circumstances.[20]. The UN Geoscheme was used in this review rather than larger continental regions (e.g. North America, Europe, etc) as it was expected that disease profiles and migration patterns may be impacted by differences in climate and environment as well as socio-economic determinants of the populations seen at this Geoscheme level. Migration routes were divided by decade based on the first year of sample collection described in the included publications: ≤1979, 1980–1989, 1990–1999, 2000–2009, 2010–2019, and ≥2020. Data synthesis was performed using Microsoft Excel and Stata/IC 16.1.

## Results

The database search retrieved 8,844 publications with an additional 11 publications identified by searching the reference lists of retrieved review papers (Fig 1). After the duplicates were removed, title and abstracts were screened for 4,295 publications. Full-texts of 604 articles were assessed for eligibility with 347 included in the final analysis.

The most frequent pathogen of interest investigated in the included literature was *Schistosoma* spp. (n = 99/347; 28.5%) followed by *Giardia lamblia* (n = 81/347, 23.3%), *Leishmania* spp. (n = 59/347, 16.4%), severe acute respiratory syndrome coronavirus 2 (SARS-CoV-2) (n = 36/347, 10.4%) and Hepatitis E virus (n = 27/347, 7.8%) (Fig 2 and S1 Table). No

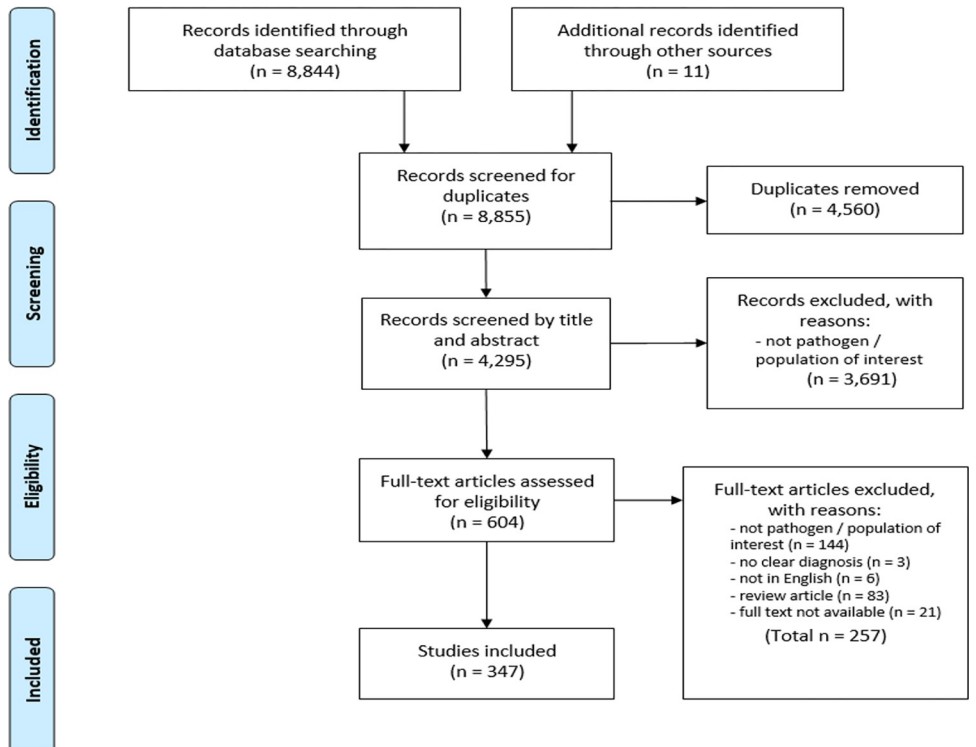

**Fig 1. Flow diagram of the studies identified, screened, reviewed, and included in the scooping review.**

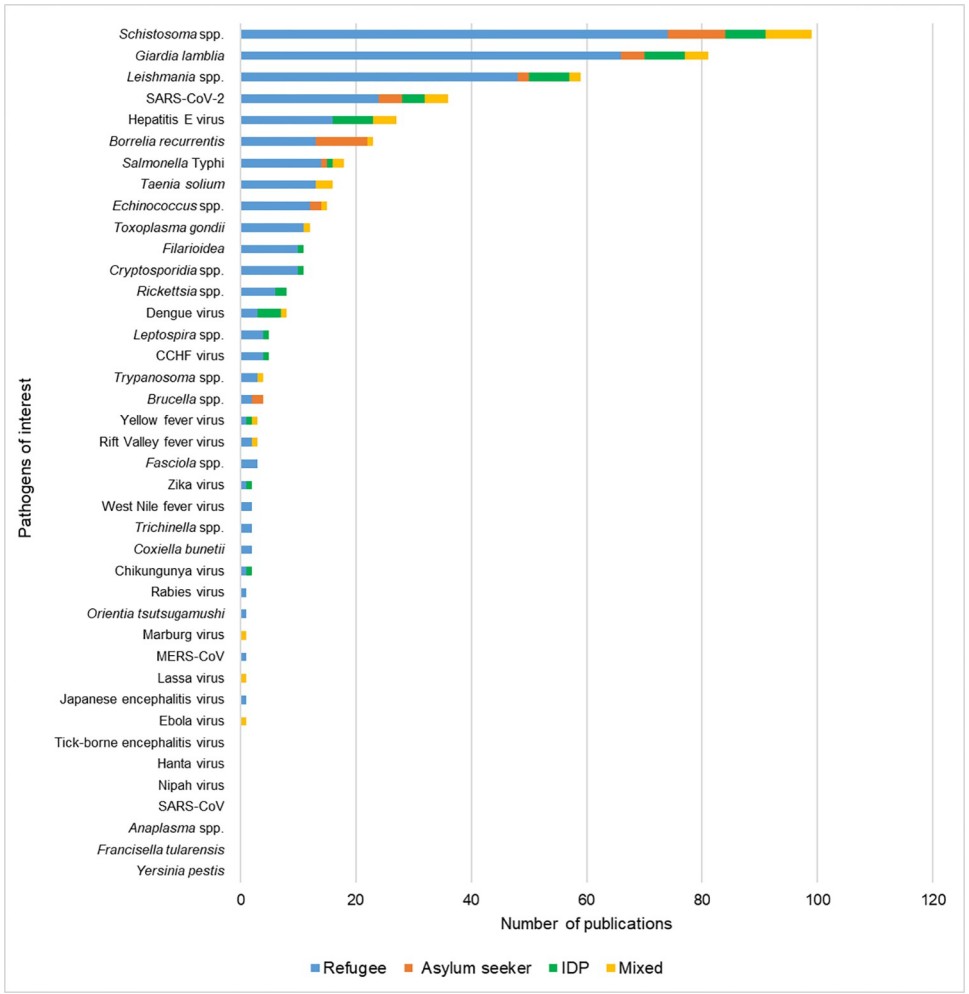

**Fig 2. Number of publications investigating zoonotic pathogens of interest in refugees, internally displaced persons and asylum seekers.** Abbreviated pathogens: severe acute respiratory syndrome coronavirus (SARS-CoV), severe acute respiratory syndrome coronavirus 2 (SARS-CoV-2), *Crimean-Congo haemorrhagic fever* (*CCHF*), *Middle East respiratory syndrome coronavirus* (*MERS-CoV*) and *Salmonella enterica* subsp. *enterica* serovar Typhi (*Salmonella* Typhi).

publications were retrieved for seven of the pathogens of interest: *Yersinia pestis*, severe acute respiratory syndrome coronavirus (SARS-CoV), Nipah virus, Hantavirus, Tick-borne encephalitis virus, *Francisella tularensis* and *Anaplasma* spp.

Negative results were reported in 18/347 (5.2%) of included publications, for at least one of the populations investigated (S2 Table). These publications looked at *Schistosoma* spp., *Filarioidea*, *G. lamblia*, *Leptospira* spp., *Salmonella* Typhi, *Echinococcus* spp., SARS-CoV-2, Hepatitis E virus, *Middle East respiratory syndrome coronavirus* (*MERS-CoV*) virus, Zika virus, Chikungunya virus, *Crimean-Congo haemorrhagic fever* (*CCHF*) virus, *Rift Valley fever* virus and Yellow Fever virus.

In addition to the zoonotic pathogens of interest, other pathogens (not included in this review) were the most frequently reported health concern in displaced people in the retrieved publicatons (n = 175/347; 50.4%) (S3 Table). These included other helminth or protozoa infections (n = 68/175; 38.9%), tuberculosis (n = 27/175; 15.4%), malaria (n = 21/175; 12.0%), HIV (n = 14/175; 8.0%) and other pathogens (n = 45/175; 25.7%). Non-communicable diseases

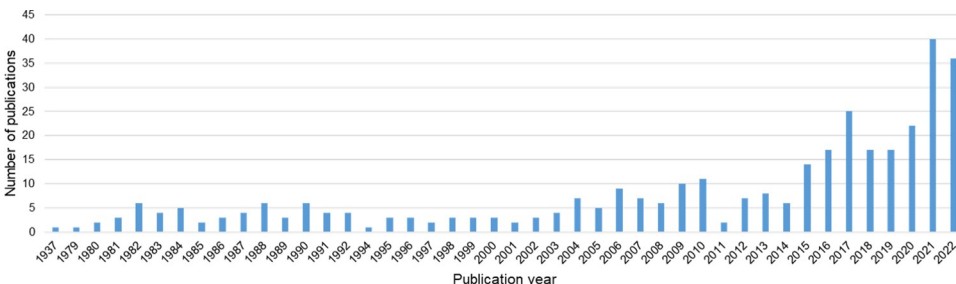

**Fig 3. Number of publications investigating zoonotic pathogens of interest in refugees, internally displaced persons and asylum seekers per year from 1937–2022.**

were mentioned in 51/347 (14.7%) publications, while malnutrion and underweight were reported in 18/347 (5.2%) publications and 4/347 (1.2%) listed overweight or obestity. Treatment was mentioned in 160/347 (46.1%) of the publications with medication (n = 148/160; 92.5%) being the most common and only a few studies mentioning surgery (n = 14/160; 8.8%), supportive care (n = 14/160; 8.8%) or other (n = 10/160; 6.3%).

Refugee populations were the most commonly investigated with 75.2% (n = 261/347) of publications, while only 11.2% (n = 39/347) and 8.4% (n = 29/347) reported on IDPs and AS, respectively (S4 Table). A further 5.2% (n = 18/347) of publications reported on mixed populations of displaced people. Displaced populations residing in camps or reception centres were the most studied, acounting for 42.7% (n = 148/347) of publications, although 30.8% (n = 107/ 347) of publications did not specifiy the setting (S5 Table). Displaced populations living within host communities were studied in 20.5% (n = 71/347) of publications and 6.1% (n = 21/347) looked at mixed settings of both camps and communities. Most of the included publications investigated "majority adults" (n = 118/347; 34.0%) or "all age ranges" (n = 108/347; 31.1%), while 38/347 (23.9%) investigated "majority children" and 38/347 (24.0%) did not specify the age of the population. The sex distribution of participants investigated in the included publications favored males with "majority male" accounting for 34.0% (n = 118/347) of publications and "equal" accounting for 32.9% (n = 114/347), while only 12.1% (n = 42/347) focusing on "majority female". Seventy-three publications did not specifiy the sex of participants.

Since the early 2000's there has been an increase in the number of publications reporting on zoonotic diseases in displaced poulations, with a sharp rise in 2021–2022 in response to the SARS-CoV-2 pandemic (Fig 3). Publications investigating SARS-CoV-2 accounted for 42.5% (n = 17/40) and 41.7% (n = 15/36) of included publications in 2021 and 2022, respectively.

Cross-sectional studies were the most common study types (n = 95/347, 27.4%) reporting on zoonotic pathogens in refugees, IDPs and AS (Fig 4; S6 Table). Followed by surveillance or screening studies (n = 77/347, 22.2%), case studies (n = 63/347, 18.2%) and outbreak reports (n = 31/347, 8.9%). There were very few diagnostic test validation studies (n = 7/347, 2.0%) or clinical trials (n = 1/347, 0.3%) retrieved.

## Migration routes of displaced people

Migration routes of displaced populations were detailed in 316/347 (91.1%) of the included publications (Fig 5, panels A-C; Fig 6, panels A-C; S7 Table). Earlier articles reporting on displaced populations between 1937 and 1979, predominantly saw refugees from South-Eastern Asia migrating to the Americas and refugees from Eastern Europe migrating to Australia after transiting through Southern Asia, North or Eastern Africa. In the 1980's and 1990's the number of studies increased, while still reporting on South-East Asian refugees, they also started

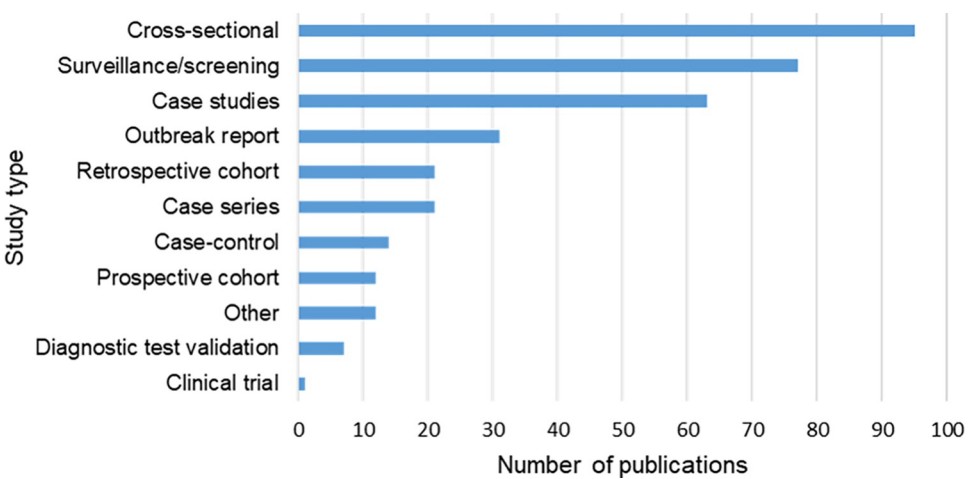

**Fig 4. Number of publications investigating zoonotic pathogens of interest in refugees, internally displaced persons and asylum seekers by study type.**

looking at refugees from Africa and South America. The 1990's saw increased attention on refugees and AS migrating to neighbouring countries within the same region as well as IDPs displaced within their own country. The 2000's saw this trend continue with the migration webs becoming more complex with displaced populations moving in both directions between some regions. Studies reporting on intra-region and intra-country migration were also found to cover more world regions. The migration routes are depicted by decade in multiple panels demonstrating the increasing complexity and trajectories of migratory pathways of humans.

## Distribution of zoonotic pathogens investigated in displaced people from Africa, Asia and Latin America

Included publications that linked a specific pathogen of interest with a displaced person or group of people from a specific origin region (n = 262/347; 75.5%) were grouped to determine the most commonly investigated pathogen in displaced people from Africa (n = 111/262; 42.4%), Asia (n = 135/262; 51.5%) and Latin America (n = 11/262; 4.2%) (Fig 7 and S8 Table). Africa included Northern, Western, Middle, Eastern and Southern Africa; Asia included Central, Western, Eastern, Southern and South-Eastern Asia; and Latin America included the Caribbean, Central and South America.

Studies in Asia investigated the most diverse number of pathogens of interest (n = 23/40; 57.5%), compared to Africa (n = 20/40; 50%) and Latin America (n = 10/40; 25.0%). In Asia the most commonly investigated pathogen of interest associated with displaced people was *Leishmania* spp. (n = 44/135; 32.6%), followed by *G. lamblia* (n = 35/135; 25.9%), *Schistosoma* spp. (n = 14/135; 10.4%) and *Echinococcus* spp. (n = 10/135; 7.4%). Conversely, *Schistosoma* spp. is the most commonly reported pathogen in displaced people coming from Africa with 46 (n = 46/111; 41.4%) publications, followed by *Borrelia recurrentis* (n = 19/111; 17.1%), Hepatitis E virus (n = 15/111; 13.5%) and *G. lamblia* (n = 13/111; 11.7%).

Very few papers (n = 11/262; 4.2%) were retrieved where a pathogen of interest was directly linked to displaced people originating in Latin America. SARS-CoV-2 was investigated the most in these populations with four publications (n = 4/11; 36.4%). Additionally, *Leishmania* spp., *G. lamblia*, Hepatitis E virus, Zika virus, Chikungunya virus, Yellow Fever virus, Dengue virus, *Trypanosoma* spp. and *Toxoplasma gondii* were each investigated in one publication

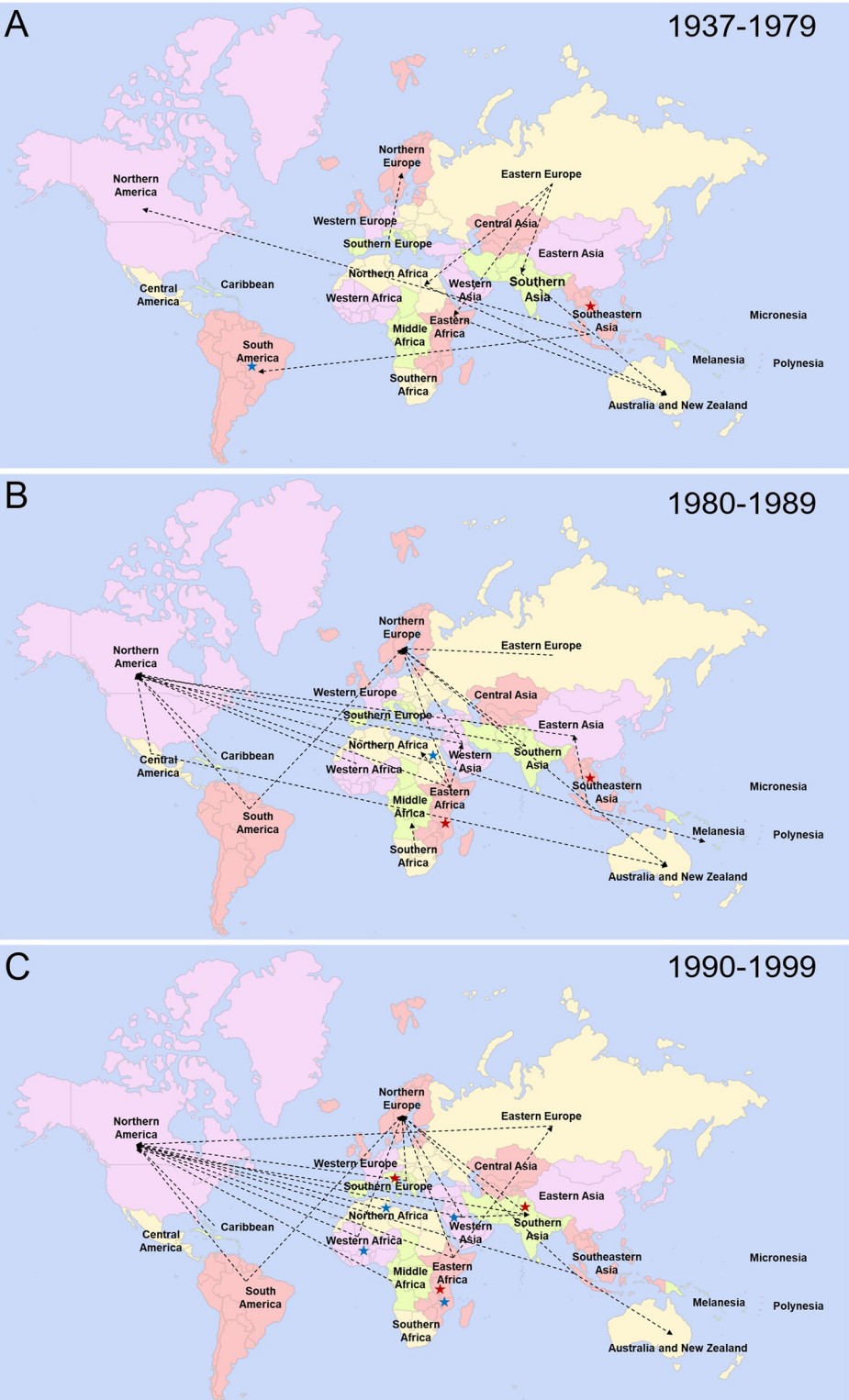

**Fig 5. Maps of migration pathways of displaced people included in this review.** Time periods were based on the first year of sample collection in each study. A) 1937 to 1979; B) 1980–1989, C) 1990–1999. Black arrows indicate the direction of migration–an arrow was included if at least one publication with sample collection initiated during the time period described the migration pathway. Blue stars mark regions with internally displace people and red stars mark regions with refugees or asylum seekers migrating between countries within the same region. Created from Fla-shop.com [21].

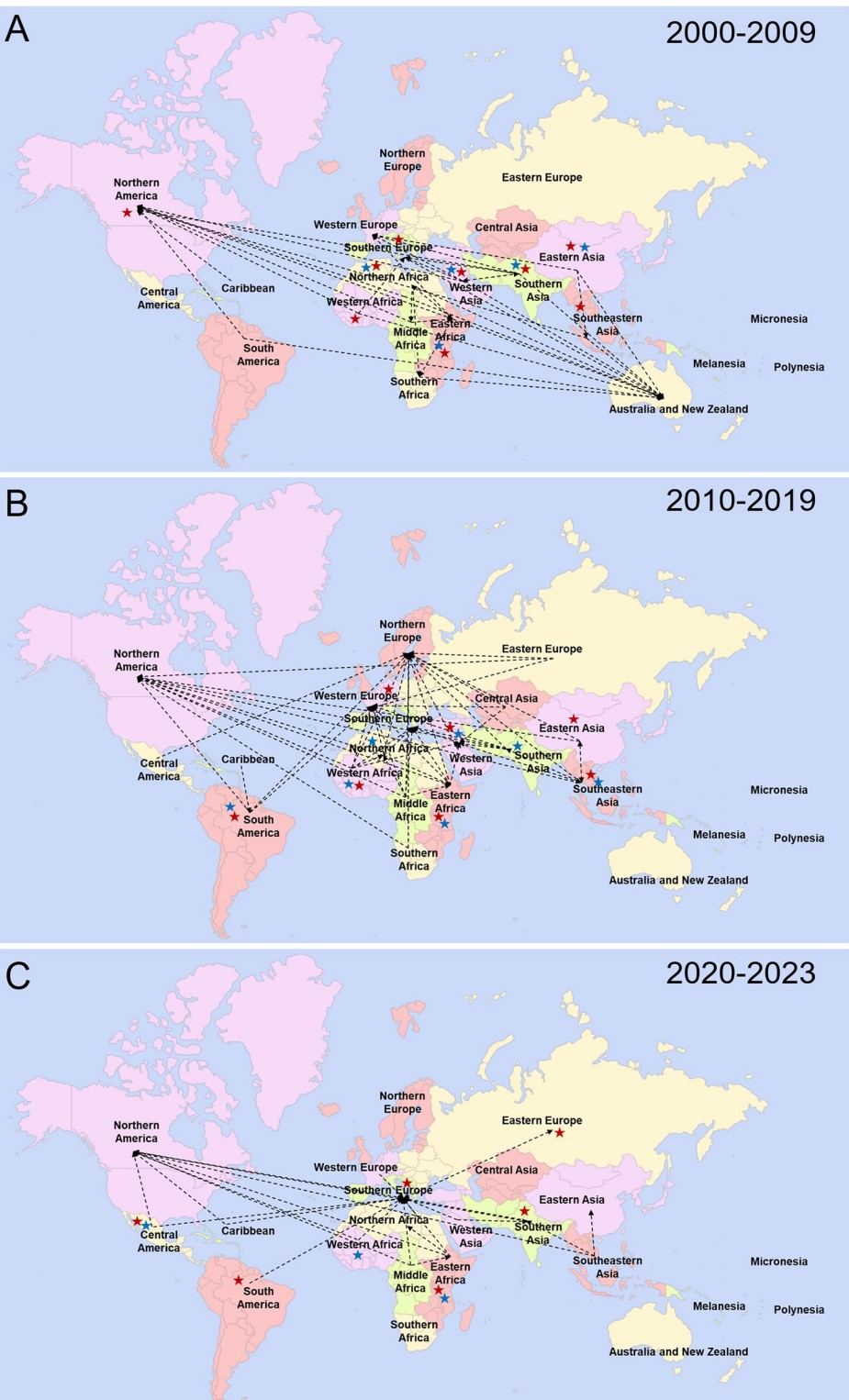

**Fig 6. Maps of migration pathways of displaced people included in this review.** Time periods were based on the first year of sample collection in each study. A) 2000–2009, B) 2010–2019, C) 2020–2023. Black arrows indicate the direction of migration–an arrow was included if at least one publication with sample collection initiated during the time period described the migration pathway. Blue stars mark regions with internally displace people and red stars mark regions with refugees or asylum seekers migrating between countries within the same region. Created from Fla-shop.com [21].

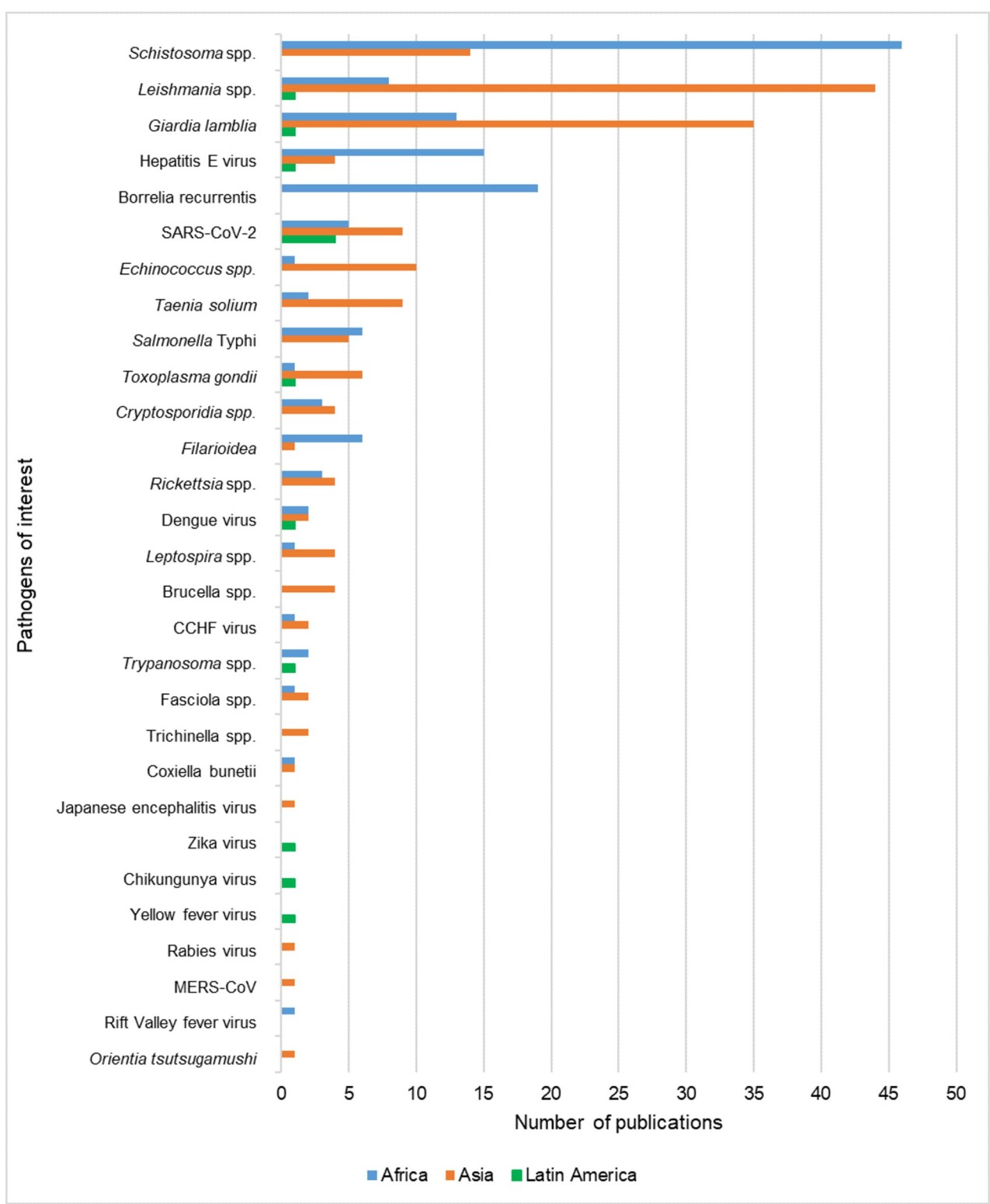

**Fig 7. Number of publications for each zoonotic pathogen of interest in three global regions: Africa, Asia and Latin America.** Abbreviated pathogens: severe acute respiratory syndrome coronavirus 2 (SARS-CoV-2), Crimean-Congo haemorrhagic fever (CCHF), Middle East respiratory syndrome coronavirus (MERS-CoV) and *Salmonella enterica* subsp. *enterica* serovar Typhi (*Salmonella* Typhi).

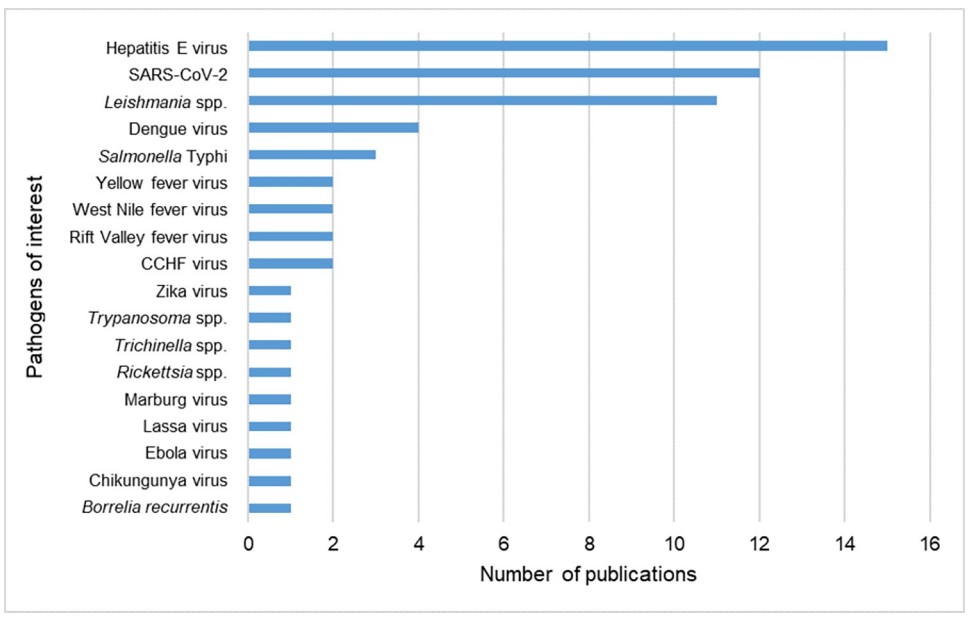

**Fig 8. Number of publications reporting zoonotic pathogens amongst displaced people in disease outbreak situations.** Abbreviated pathogens: severe acute respiratory syndrome coronavirus 2 (SARS-CoV-2), Crimean-Congo haemorrhagic fever (CCHF) and *Salmonella enterica* subsp. *enterica* serovar Typhi (*Salmonella* Typhi).

(n = 1/11; 9.1%). None of the other pathogens of interest were found with a direct link to a displaced person or people originating in Latin America.

## Zoonotic pathogens associated with disease outbreaks in displaced people

Disease outbreaks related to the pathogens of interest were described in 46 (n = 46/347, 13.3%) publications, 71.7% (n = 33/46) of these occurred in camp/reception centre settings. Outbreaks were associated with 18/40 (45%) of the pathogens of interest included in this review, of which 12/18 (66.7%) were viruses (Fig 8 and S9 Table). The most commonly reported pathogens of interest in association with outbreaks was Hepatitis E virus (n = 17/46, 37.0%), followed by SARS-CoV-2 (n = 12/46, 26.1%) and *Leishmania* spp. (n = 11/46, 23.9%), with Dengue virus (n = 4/46, 8.7%) and *Salmonella* Typhi (n = 3/46, 6.5%) making up for the top five most published pathogens of interest associated with displaced populations and disease outbreaks.

## Predictors of infection with zoonotic pathogens in displaced people

Predictors of infection with a pathogen of interest found to be statistically significant (confirmed) were reported in 25.6% (n = 89/347) of the publications retrieved, while in 53.9% (n = 187/347) of the publications the authors discussed potential predictors of infection (presumptive; not tested for statistically) (Fig 9 and S10 Table). The most frequently identified predictors of infection were "vulnerable groups" i.e. women, children, ethnic groups etc. (n = 32/89, 36.0%), "refugee status" (n = 17/89, 19.1%), and "travel to or from endemic regions" (n = 12/89, 13.5%). When combined with the suggested exposures, however, limited access to hygiene and sanitation and crowding were the most commonly discussed. While limited access to healthcare and delayed or mis-diagnosis were only identified in three publications each, they were suggested in a further 35/187 (18.7%) and 18/187 (9.6%) publications, respectively. Contact with animals or vectors were identified in 6/89 (6.7%) publications and suggested in an additional 16/187 (8.6%). Environmental factors included climate or climate change and

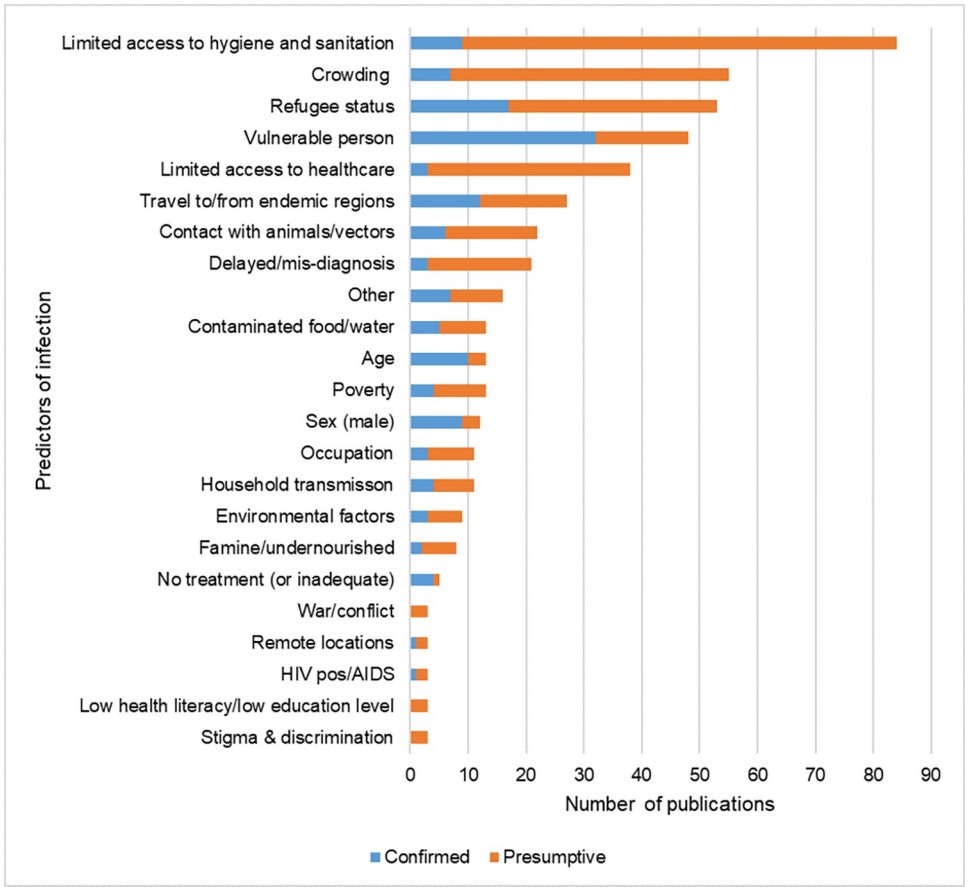

**Fig 9. Number of publications reporting predictors of infection with zoonotic pathogens in displaced people.**
Confirmed predictors of infection were those found to be statistically significant. Presumptive predictors of infection were suggested by the authors of the included publications.

periods of flooding while additional predictors of infection characterized as "other" included factors such as lack of mosquito net use, seasonality, other pre-existing conditions, substance abuse, etc.

## Preventative measures against infections and transmission of zoonotic pathogens

Preventative measures were discussed by the authors of 28.5% (n = 99/347) of the publications retrieved (Fig 10 and S11 Table). The most common preventative measures mentioned were vaccines or prophylaxis medications including mass drug administration (MDA) campaigns (n = 38/99; 38.4%) and surveillance programs (n = 35/99; 35.4%). Improving hygiene and sanitation facilities were also common with 28/99 (28.3%) publications. Animal and vector control programs were included in 20/99 (20.2%) publications while healthcare worker education and improved access to healthcare were discussed in 8/99 (8.1%) and 5/99 (5.1%) publications, respectively.

## Discussion

This scoping review on clinically important zoonotic pathogens aimed to determine the state of global zoonotic disease research in refugees, AS and IDPs. A total of 347 articles were

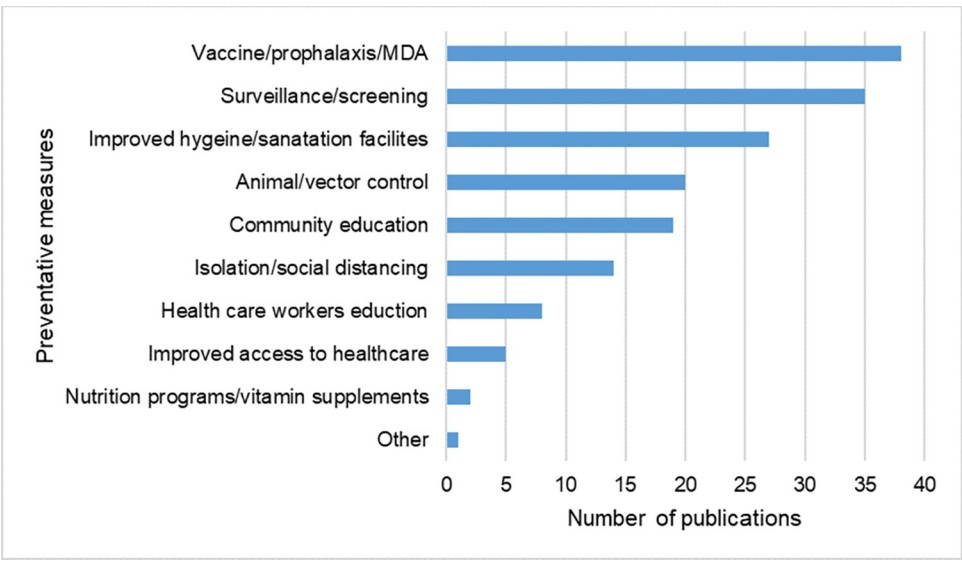

**Fig 10. Number of publications reporting preventative measures against zoonotic infections in displaced people.** Abbreviated term: mass drug administration (MDA).

included in the review, and it is apparent that there are important gaps in the available evidence describing zoonotic pathogens and their impact, including identification of risk factors and transmission pathways, for displaced populations.

## Zoonoses research in displaced populations

Available research reporting on a number of the zoonotic pathogens of interest was lacking, with seven retrieving no records at all, including; three bacteria (*Y. pestis*, *F. tularensis*, *Anaplasma* spp.) and four viruses (SARS-CoV, Nipah virus, Hantavirus and Tick-borne encephalitis virus). Further, only 30% (n = 12/40) of the pathogens of interest were studied in more than 10 publications and only 3/40 (7.5%) were found in more than 40 publications. These findings highlight an important research gap in the current literature, but may also reflect the (perceived) relevance of these pathogens on displaced people, or a publication bias. Lack of relevance as an explanation for the low number of publications retrieved for the pathogens of interest is unlikely, with the WHO publishing a report in 2012 stressing the importance of neglected zoonoses in marginalized populations for over 10 years since [22]. Challenging and expensive diagnostics or poor recognition due to non-specific symptoms or unfamiliarity in non-endemic regions may lead to pathogens being underreported. *Y. pestis* for example often evokes reflections on a historical plague, however, the outbreak of pneumonic plague in Madagascar in 2017 with 2,414 suspected cases highlights that it remains an important disease [23]. A crowded, unhygienic camp setting has the potential to promote outbreaks with transmission by fleas with rodent reservoirs in addition to human-to-human transmission possible by respiratory droplets [24]. In settings with limited access to healthcare and antibiotics, the disease may spread and manifest rapidly and often with fatal outcomes [24]. Molecular diagnostics or bacterial culture is recommended, although misidentification may occur, requiring confirmation by a reference laboratory [24]. *Anaplasma* spp. are obligate intracellular bacteria, requiring sophisticated laboratories to perform polymerase chain reaction (PCR) assays, cell culture or skilled microscopists for detection in indirect immunofluorescence assays (IFAs) or in blood smears [25]. Both *Anaplasma* spp. and *F. tularensis* infections are difficult to distinguish

clinically due to non-specific symptoms [25,26]. Viruses, typically diagnosed by molecular or serological methods, also require sophisticated laboratories [27–29]. Nipah virus, while associated with a number of outbreaks in Asia, is less recognized in other regions of the world and may be missed by clinicians treating migrating people [27]. Publication bias for positive results may also play part in the limited number of publications retrieved with only 18/347 (5.2%) reporting negative results. Despite the scientific community recognizing the importance of negative results they are still less likely to be published.

In contrast, the three most frequently studied zoonotic pathogens of interest in displaced populations globally are *Schistosoma* spp., *G. lamblia* and *Leishmania* spp. These pathogens require simple inexpensive diagnostics, such as microscopy or clinical examination [30–33]. Increasing awareness and the development of affordable and easy to use diagnostics would allow for a greater breadth of pathogens to be investigated in vulnerable populations including displaced people, in both the clinical and research setting. A recent review revealed, that almost two-thirds of adolescent and adult febrile patients attending health care facilities in East Africa might receive inappropriate management, partly due to diagnostic limitations associated with missed or wrong identification of fever aetiology [34].

Interest in zoonotic diseases appears to be increasing, with the number of publications rising since the early 2000's, although this may reflect in an increase in research in general (Figs 5 and 6). The SARS-CoV-2 pandemic saw a sharp rise in the number of publications, however, it remains to be seen if this will translate into an increased interest in research for other more neglected zoonotic pathogens.

Research on zoonotic pathogens in displaced populations focuses primarily on refugees, with AS and IDPs being comparatively neglected. This trend is also observed in the general healthcare needs of AS and IDPs, who often do not have the same international legal protection as refugees and do not receive as much attention or support from the international community [35,36]. IDPs in particular are generally located in LMICs and migrate only short distances, meaning they are often still impacted by the risks associated with conflict [36]. The feasibility of conducting research in IDP and AS populations can also be affected by these challenging conditions and may explain the disparity seen with research in refugees, some of who will have reached more stable settings and undergone pre-departure or post-arrival screening. The UN International Organization for Migration (IOM) has 175 member states and provides pre-migration health screening (PMHS) services [37,38]. Currently, the host country tailors their PMHS approaches to their entry requirements (physical or mental health screening, vaccines, linkage to care etc.). Examples of countries that use the IOM PMHS services are New Zealand and Canada. The "Immigration and Refugee Health Working Group" aims to assess and promote standardised international best practices for IOM PMHS services (Australia, Canada, New Zealand, the United Kingdom and the United States of America) [37]. In New Zealand, post-arrival medical screening is provided to refugees at the resettlement centres where they are housed on arrival [39]. Contrary to other settings where refugees are directly housed in the community and patients may first need to access primary health services to obtain medical screening [39].

## Zoonotic research in displaced populations from Africa, Asia and Latin America

Different patterns of research focus were observed based on the region of origin of the study participants. Investigations of displaced people from Africa focused on *Schistosoma* spp., followed by *B. recurrentis* and Hepatitis E virus. In Asia, *Leishmania* spp. takes the top place followed by *G. lamblia* while *Schistosoma* spp. is the third most commonly investigated pathogen of interest.

Displaced people originating from Latin America were the most under-represented in the publications retrieved—with only 11 out of the 262 papers directly linking a displaced person with their country of origin and a pathogen of interest. While this may in part be due to a language bias with only English publications included in the review, the extremely low number of publications is still suggestive of a substantial research gap in displaced populations from Latin America. These papers included 4/11 (36.4%) responding to the SARS-CoV-2 pandemic, while the remaining six publications cover only nine of the pathogens of interest included in this review. Historically, the number of refugees from the Americas (6.1 million; between 1975 and 2022) has been lower than other regions of the world with 27.2 million from Sub-Saharan Africa, 15.7 million from Asia and the Pacific and 11.7 million from the Middle East and North Africa, during the same time period [1]. This may in part account for the lower number of papers on displaced people originating from Latin America. This region is growing in importance for displaced populations with six of the top ten source countries for new asylum applications ((Venezuela, Afghanistan, Cuba, Nicaragua, Ukraine, Syria, Colombia, Honduras, Haiti and Türkiye; 2021 and 2022) [1]. Of the 89.3 million forcibly displaced people globally as of the end of 2021, 4.4 million were Venezuelans–representing only one group of displaced people in Latin America [40]. This review highlights the need for further research into health status and needs of these displaced populations.

## Outbreak associated factors

"Limited access to hygiene" and "sanitation" and "crowding" were the most commonly discussed predictors for infection in the displaced populations included in this review. Additionally, residing in a "camp" or "reception centre" settings were common, and these predictors are commonly associated to the development of outbreaks [41]. Viruses were the most reported pathogen type in association with outbreaks, accounting for 14/18 (77.8%) pathogens associated with outbreaks investigated in this review. However, only half of the viruses included in this review were investigated in more than one publication. Furthermore, viruses accounted for four of the seven pathogens that were not investigated in the retrieved literature on displaced people. These findings may reflect the more complex conventional diagnostics required for virus detection compared to bacteria and parasites, and indicate a gap in the research for outbreak prevention in displaced populations.

## Predictors of infection

A large number of predictors of infection were identified (confirmed) and suggested (presumptive) in the retrieved publications. Interestingly, the predictors most often suggested (limited access to hygiene and sanitation, crowding and limited access to healthcare) were not often tested for statistically, reflecting the challenge of discerning the impact specific factors have on these vulnerable populations. The predictors discussed are generally not independent factors but are rather an inter-connected web that proliferate and intensify each other. Displacement leads to increased poverty through loss of assets and/or employment [42]. Poverty limits displaced peoples' access to safe drinking water and food as well as the ability to pay for healthcare. Malnutrition and unsafe drinking water increases the risk of infectious disease. Similarly, displacement interrupts education leading to lower education and health literacy levels [42]. Displaced people may be housed in crowded, makeshift settlements without sufficient hygiene and sanitation, increasing the risk of outbreaks [43]. Movement of people to new regions where healthcare workers may not be familiar with pathogens from their place of origin can delay or cause mis-diagnosis [44,45]. Conversely, immunologically naive populations may become exposed to new pathogens in their host regions.

Large proportions of forcibly displaced people migrate due to conflict in their place of origin. Conflict has been shown to not only increase poverty-related poor health but also be associated with increased disease prevalence in its own right [42]. Conflict may cause health workers to migrate, be injured or killed. Conflict can also cause damage to healthcare or supporting infrastructure (e.g. power or water agencies); disrupt transport, supply chains and communication; and disrupt public health programs (e.g. vector control programs, MDA) [42]. Furthermore, conflict may damage sanitation services (sewage treatment, rubbish collection and disposal) or the environment, increasing the risk of unsafe drinking water and/or increasing the number of vectors and animal reservoirs (e.g. rodents) or altering their distribution leading to greater transmission of infectious diseases [42].

Although not addressed in this review, the movement of companion animals and livestock with displaced people should also be considered. Rohingya refugees migrating to Bangladesh brought thousands of sheep, goats, cows and buffaloes [15]. Like their human counterparts, migrating animals may introduce new pathogens to their host region, or conversely, naive animals be susceptible to pathogens present in the host or transit regions [15]. Pathogens circulating in animals presents a risk of transmission to their owners. To reduce the burden of zoonoses in displaced people, interventions would need to take a comprehensive approach in addressing these exposure factors.

## Preventative measures and the role of One Health

Vaccination and prophylactic drug administration as well as surveillance was widely discussed in the retrieved literature with early detection and prevention being essential to avoid illness in displaced populations and outbreak situations. One Health approaches including human and animal health, entomological and environmental studies are optimal to combat zoonoses. Surveillance and detection in animals, vectors or environmental reservoirs has the potential to allow public health measures to be put in place before transmission to the human population occurs [46]. Similarly, disease control in animals, such as vaccination campaigns can prevent human exposure [47].

## Limitations

A limitation of this scoping review was that we aimed to identify the state of research and literature on a selection of pathogens of interest, rather than for all known zoonoses. The aim was not to represent all clinically relevant zoonotic pathogens in this review (e.g. bovine tuberculosis), and a limitation is that we could not therefore define the most commonly studied pathogens of all zoonoses. This was due to the feasibility of creating a database search string that would accurately include all the relevant terms for all zoonotic pathogens. Many publications reporting zoonotic diseases consider only the human element–particularly in terms of migration health. Subsequently, these publications do not always include the terms "zoonoses" or "zoonotic disease" and so cannot be retrieved using these search terms alone. This review also only reflects the state of research on the selected zoonoses in displaced people, but cannot comment on the impact of the selected pathogens on these populations. This review included a wide range of study designs that cannot be directly compared, additionally; the specific study types cannot be directly compared due different inclusion/exclusion criteria and testing methods. This review included only peer-reviewed published original articles in English. Not including publications available only in Spanish and Portuguese may have affected the number of publications reporting on displaced populations from Latin America and possibly also Africa. The exclusion of French literature may also have impacted the number of publications retrieved. A large number of the retrieved publications either did not specify the region of

origin for study participants or listed a number of regions but did not specify which displaced population were infected by the pathogens of interest. Subsequently, the data synthesis on patterns of research in displaced people from different geographical regions only included a subset of the retrieved papers. Similarly, the migration routes only reflect the displaced populations from publications that included a full migration pathway and sampling year(s). Migration routes of pathogens could not be addressed as the data did not distinguish whether the infection was acquired in the study participant's origin, transit or host country. The predictors of infection and preventative measures discussed were generally not linked to specific pathogens of interest in the original articles, which often also included other health concerns, as such we can only describe them generally as to how they may apply to zoonoses in displaced populations.

## Conclusions

Displaced populations are susceptible to poor health and infectious diseases, with those residing in crowded and often unhygienic camp settings being particularly vulnerable to disease outbreaks. Despite the risk zoonoses pose to human health and their potential to cause outbreaks, research is focused on a limited number of pathogens. To fully ascertain the role zoonoses have in the health of displaced people, research should be expanded to cover a wider selection of diseases, particularly those with epidemic potential and include animal hosts accompanying displaced people. Greater focus on AS and IDPs is also required, being particularly neglected in comparison to refugee populations. These gaps could be addressed by increased awareness in both clinical and research environments as well as improved diagnostics suited for use in settings where full laboratories may not be available (e.g. accurate and inexpensive RDTs–for humans and animals). Furthermore, One Health studies are best placed to comprehensively address zoonoses and their transmission routes, to subsequently guide public health measures to reduce their transmission and impact in these vulnerable populations.

## Supporting information

**S1 Text. PRISMA-ScR Checklist.**
(DOCX)

**S2 Text. Pubmed search string.**
(DOCX)

**S1 Table. Publications included in the scoping review.**
(DOCX)

**S2 Table. Publications included in the scoping review reporting negative results.**
(DOCX)

**S3 Table. Publications included in the scoping review reporting other health concerns.**
(DOCX)

**S4 Table. Publications included in the scoping review reporting on refugee, asylum seekers, internally displaced people or mixed populations.**
(DOCX)

**S5 Table. Publications included in the scoping review reporting on displaced people in camp/reception centre, community or mixed settings.**
(DOCX)

**S6 Table. Publications included in the scoping review reporting by study type.**
(DOCX)

**S7 Table. Publications included in the scoping review reporting migration routes of displaced people, by first year of sample collection.**
(DOCX)

**S8 Table. Publications included in the scoping review reporting on zoonotic pathogens in displaced people from Africa, Asia and Latin America.**
(DOCX)

**S9 Table. Publications included in the scoping review reporting on zoonotic pathogens associated with disease outbreaks.**
(DOCX)

**S10 Table. Publications included in the scoping review reporting on predictors of infections.**
(DOCX)

**S11 Table. Publications included in the scoping review reporting on preventative measures against infections and transmission of zoonotic pathogens.**
(DOCX)

**S12 Table. Data extraction.**
(XLSX)

## Acknowledgments

We thank Christian Appenzeller-Herzog from the University Medical Library, University of Basel, for his support in reviewing the search strategy.

## Author Contributions

**Conceptualization:** Regina Oakley, Rea Tschopp, Daniel H. Paris.

**Data curation:** Regina Oakley.

**Formal analysis:** Regina Oakley.

**Funding acquisition:** Daniel H. Paris.

**Investigation:** Regina Oakley, Nadja Hedrich, Alexandra Walker, Habtamu Merha Dinkita, Charles Abongomera.

**Methodology:** Regina Oakley, Rea Tschopp, Daniel H. Paris.

**Project administration:** Regina Oakley.

**Supervision:** Rea Tschopp, Charles Abongomera, Daniel H. Paris.

**Validation:** Regina Oakley, Nadja Hedrich, Alexandra Walker, Habtamu Merha Dinkita, Charles Abongomera.

**Visualization:** Regina Oakley.

**Writing – original draft:** Regina Oakley.

**Writing – review & editing:** Regina Oakley, Nadja Hedrich, Alexandra Walker, Rea Tschopp, Charles Abongomera, Daniel H. Paris.

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
