## [Decision Letter · Decision Letter 0]

15 Nov 2023

Dear Miss Oakley,

Thank you very much for submitting your manuscript "Status of zoonotic disease research in refugees, asylum seekers and internally displaced people, globally: A scoping review of forty clinically

important zoonotic pathogens" for consideration at PLOS Neglected Tropical Diseases. As with all papers reviewed by the journal, your manuscript was reviewed by members of the editorial board and by several independent reviewers. In light of the reviews (below this email), we would like to invite the resubmission of a significantly-revised version that takes into account the reviewers' comments. 

We cannot make any decision about publication until we have seen the revised manuscript and your response to the reviewers' comments. Your revised manuscript is also likely to be sent to reviewers for further evaluation.

Sincerely,

Victoria J. Brookes

Section Editor

Victoria Brookes

Section Editor

Reviewer's Responses to Questions

**Key Review Criteria Required for Acceptance?**

**Methods**

-Are the objectives of the study clearly articulated with a clear testable hypothesis stated?

-Is the study design appropriate to address the stated objectives?

-Is the population clearly described and appropriate for the hypothesis being tested?

-Is the sample size sufficient to ensure adequate power to address the hypothesis being tested?

-Were correct statistical analysis used to support conclusions?

-Are there concerns about ethical or regulatory requirements being met?

Reviewer #1: Line 127: the reference is missing, then it is not clear why the work is only about 40 zoonotic pathogens.

Line 135: reference missing, comment as above

Table 1 (page 7) maybe presenting pathogens in alphabetic order within pathogen groups (bacteria, viruses, parasites) could be appropriate (in this table it does not seem to be about relevance, occurrence)

Reviewer #2: Methodology is sound.

**Results**

-Does the analysis presented match the analysis plan?

-Are the results clearly and completely presented?

-Are the figures (Tables, Images) of sufficient quality for clarity?

Reviewer #1: Figure 1: maybe this will feet better in the method section

Line 208: check for spelling (retrieved)

Line 270: it is not clear why sample collection as in the introduction (line 128) it is said that “Only publication with a clinical or laboratory diagnosis of one of the pathogens were included”. That means that for some there were no lab diagnosis. Please check and 

Line 270-276: maybe this will fit better in the method section

Line 289-294: this can be deleted as the entire list of the 40 pathogens of interest is provided (cfr lines 310-311)

Line 300-304: same as above

Lines 330-342: it is not clear what exactly was/were “suggestive potential predictors of infection”

Figure 8: spelling (Hygiene)

Figure 8 is reported two times, please check

Line 370: spelling

Figure 9: please provide spelling for acronym

Discussion

Line 380-381: there are gap regarding impact but also regarding description, maybe this aspect could be added

Line 406: please provide spelling for IFAs (Immunofluorescence Agglutination, Immunofluorescence Assay, Indirect Fluorescent Antibody test…)

Line 423: please confirm it is “OF” or should be “OR”

Line 436:please rephrase

Line 440: Latin America instead of Africa

Line 465 & 478 (also line 338): As most of the original works were cross-sectional studies, then it is more about “associated factors as you mentioned on lines 319 320) regarding outbreaks in the results section. Please make appoint on the that in the discussion and rephrase where appropriate (risk, predictors).

Limitations

Lines 532-534: Not including publication might also had influence on the number of publications (from Africa)

Reviewer #2: Results are fine. However, I found two Figure 8 in the manuscript, so the authors have to revise the manuscript accordingly.

**Conclusions**

-Are the conclusions supported by the data presented?

-Are the limitations of analysis clearly described?

-Do the authors discuss how these data can be helpful to advance our understanding of the topic under study?

-Is public health relevance addressed?

Reviewer #1: Line 557 : spelling

Reviewer #2: Conclusions are supported by the data.

**Editorial and Data Presentation Modifications?**

Reviewer #1: Abstract

Line 36: For a reader relying only to the abstract, it may appear that there are only 40 clinically zoonotic pathogens of interest (in general), please rephrase to highlight that your work is about chosen 40 pathogens.

Line 46 : maybe it could be better to add “dates” of publications ranged from 1937 to 2022 (need to check in the text it said date of sample collection; please confirm.

Introduction

Line 87: please delete “are” after LMICs

Line 89-90: It is no clear whether 20% represent illnesses & mortality

Reviewer #2: I suggest that the manuscript undergo Major Revision before it could be considered again for publication.

**Summary and General Comments**

Reviewer #1: Thank you for the opportunity to review this manuscript. 

The manuscript by Oakley and colleagues touches an important issue: zoonotic pathogens in a forcibly displaced people (refugees, asylum seekers and internally displaced people). 

As the authors show in their manuscript, only few data exist regarding zoonotic infection in this “special population” of displaced people and for the few that exist there is a gap in the spectrum of pathogens studied. 

One of the strength is to bring attention to zoonotic infection in displaced people, knowing that more than 60% of emerging and re-emerging infections are zoonotic and that forcibly displaced people are often in crowded settings with poor hygiene and sanitation and low access to health services . These results highlight the fact that there is still need to describe the burden and the impact of zoonotic diseases in displaced population for a better planning of interventions. 

However, in the manuscript, the authors fail to be more clear in what was the rationale for choosing only 40 zoonotic pathogens. The author mentioned tuberculosis among the most reported infection in human (line 210) not included in this review. One could hope to see data on zoonotic tuberculosis (tuberculosis/bovine tuberculosis) and other zoonotic infections; or a clear explanation why the restricted 40 pathogens only.

Reviewer #2: It will be great if the authors elaborate on the differences between refugees, asylum seekers and internally displaced people, since these respective cohorts are central to the story of the manuscript. I particularly enjoyed reading the authors describing an example of the Rohingya refugees migrating to Bangladesh with their livestock to show the risk of disease transmission (Lines 504-506). I believe the same should be described for refugees, asylum seekers and internally displaced people, so that the manuscript is further strengthened.

Does the study require registration with the IRB? I suggest the manuscript to be sent for language editing.

Several revisions to be made, see below;

Table 1 - replace MERS virus with MERS CoV, replace SARS virus with SARS CoV and Covid with coronavirus disease (COVID-19)

Line 190 - Hanta virus or hantavirus?

Line 196 - severe acute respiratory syndrome coronavirus (SARS CoV). Please standardize throughout.

Line 197 - Middle East Respiratory Syndrome coronavirus (MERS CoV). Please standardize throughout.

Line 284-286 - Geographical terms eg. Northern/Western/Eastern should probably be checked for accuracy. The commonly accepted terms are North Africa/West Africa/Southeast Asia etc.

Line 292 & 302 - chikungunya

Line 293 & 303 - West Nile virus

Fig 8 - There are two Fig 8 in the document.

PLOS authors have the option to publish the peer review history of their article (what does this mean?). If published, this will include your full peer review and any attached files.

Reviewer #1: No

Reviewer #2: No
---

## [Decision Letter · Decision Letter 1]

14 Mar 2024

Dear Dr Oakley,

Thank you very much for submitting your manuscript "Status of zoonotic disease research in refugees, asylum seekers and internally displaced people, globally: A scoping review of forty clinically

important zoonotic pathogens" for consideration at PLOS Neglected Tropical Diseases. As with all papers reviewed by the journal, your manuscript was reviewed by members of the editorial board and by several independent reviewers. In light of the reviews (below this email), we would like to invite the resubmission of a significantly-revised version that takes into account the reviewers' comments. 

We cannot make any decision about publication until we have seen the revised manuscript and your response to the reviewers' comments. Your revised manuscript is also likely to be sent to reviewers for further evaluation.

Sincerely,

Victoria J. Brookes

Section Editor

Victoria Brookes

Section Editor

Reviewer's Responses to Questions

**Key Review Criteria Required for Acceptance?**

**Methods**

-Are the objectives of the study clearly articulated with a clear testable hypothesis stated?

-Is the study design appropriate to address the stated objectives?

-Is the population clearly described and appropriate for the hypothesis being tested?

-Is the sample size sufficient to ensure adequate power to address the hypothesis being tested?

-Were correct statistical analysis used to support conclusions?

-Are there concerns about ethical or regulatory requirements being met?

Reviewer #2: OK

**Results**

-Does the analysis presented match the analysis plan?

-Are the results clearly and completely presented?

-Are the figures (Tables, Images) of sufficient quality for clarity?

Reviewer #2: OK

**Conclusions**

-Are the conclusions supported by the data presented?

-Are the limitations of analysis clearly described?

-Do the authors discuss how these data can be helpful to advance our understanding of the topic under study?

-Is public health relevance addressed?

Reviewer #2: OK

**Editorial and Data Presentation Modifications?**

Reviewer #2: (No Response)

**Summary and General Comments**

Reviewer #2: The authors have revised the manuscript according to my earlier comments, however I still find that the manuscript could be further improved. Please refer to my comments below;

Line 414-hantavirus (please revise)

Lines 462-466-The feasibility of conducting research in IDP and AS populations can also be affected by these challenging conditions and may explain the disparity seen with research in refugees, some of who will have reached more stable settings and undergone pre-departure or post-arrival screening (Can the authors give an example of any country that has such health screening initiative for refugees? Perhaps even discuss the role of UNHCR?)

Figure 5-The descriptions of the respective continents/regions have to be corrected. Eg. North America, Southeast Asia, West Africa, Central Africa; not Northern America, Southeastern Asia, Western Africa, Middle Africa. It is not just limited to the above mentioned regions, so please revise them accordingly. Also, please describe New Zealand separately.

PLOS authors have the option to publish the peer review history of their article (what does this mean?). If published, this will include your full peer review and any attached files.

Reviewer #2: No
---

## [Editor Report · Decision Letter 2]

23 Apr 2024

Dear Dr Oakley,

We are pleased to inform you that your manuscript 'Status of zoonotic disease research in refugees, asylum seekers and internally displaced people, globally: A scoping review of forty clinically

important zoonotic pathogens' has been provisionally accepted for publication in PLOS Neglected Tropical Diseases.

Best regards,

Victoria J. Brookes

Section Editor

---

## [Editor Report · Acceptance letter]

6 May 2024

Dear Dr Oakley,

We are delighted to inform you that your manuscript, "Status of zoonotic disease research in refugees, asylum seekers and internally displaced people, globally: A scoping review of forty clinically
important zoonotic pathogens," has been formally accepted for publication in PLOS Neglected Tropical Diseases.

Best regards,

Shaden Kamhawi

co-Editor-in-Chief

Paul Brindley

co-Editor-in-Chief
